



# Technical Note: Evaluation of the Skill in Monthly-to-Seasonal Soil Moisture Forecasting Based on SMAP Satellite Observations over the Southeast US

Amirhossein Mazrooei[1], A. Sankarasubramanian[1], and Venkat Lakshmi[2]

[1]Department of Civil, Construction, and Environmental Engineering, North Carolina State University, Raleigh, North Carolina
[2]Department of Engineering Systems and Environment, University of Virginia, Charlottesville, Virginia

**Correspondence:** Amir Mazrooei (amazroo@ncsu.edu)

**Abstract.** Providing accurate soil moisture (SM) conditions is a critical step in model initialization in weather forecasting, agricultural planning, and water resources management. This study develops monthly to seasonal (M2S) top layer SM forecasts by forcing 1-3 month ahead precipitation forecasts with Noah3.2 Land Surface Model. The SM forecasts are developed over the Southeast US (SEUS) and the SM forecasting skill is evaluated in comparison with the remotely sensed SM observations collected by Soil Moisture Active Passive (SMAP) satellite. Our results indicate potential in developing real-time SM forecasts. The retrospective 18-months (April 2015 - September 2016) comparison between SM forecasts and the SMAP observations shows statistically significant correlations of 0.62, 0.57, and 0.58 over 1-3 month lead times respectively. As a case study, the evaluation of the issued forecasts based on the drought indexes monitored during the 2007 historical drought over the SEUS also indicate promising skill in monthly SM forecasting to support agricultural planning and water management for such natural hazards.

## 1 Introduction

Seasonal climate forecasts provide beneficial information for developing hydrologic forecasts that support planning and management of water resources. Likewise, accurate soil moisture (SM) forecasting can significantly assist the decision making for agricultural systems. Most evaluation of climate forecasts has traditionally focused only on the skill in predicting seasonal precipitation, temperature and the resultant teresterial fluxes, primalily monthly-to-seasonal streamflow (Devineni et al., 2008; Armal et al., 2018; Mazrooei et al., 2015) Also, studies have focused on the utility of climate forecasts for agriculture systems by evaluating the skill in predicting seasonal crop yield under rain-fed agriculture (Hansen et al., 2006). As rain-fed agriculture heavily depends on actual soil moisture conditions and the stress that crops face during the growing phase, long-range SM forecasts would be more advantageous to improve crop yield forecasts. Moreover, accurate prediction of initial hydrologic conditions (IHC) enhances the estimation of land surface feedback to the atmosphere in regional climate models and successively enhances the skill in seasonal hydrologic forecasts (Koster and Suarez, 2001; Berger and Entekhabi, 2001; Wood et al., 2002).



Most efforts to develop SM forecasts through land-surface models (LSMs) have actually been compared with the simulated SM from LSMs using observation-based atmospheric forcings (Mo et al., 2012; Mo and Lettenmaier, 2014). Nevertheless, systematic evaluation of our ability to forecast actual SM has not been carried out due to the limited availability of high quality observed SM data over large domains. Thus, comparison of SM forecasts with remotely sensed SM observations holds

a considerable potential. Remote sensing of SM observations using microwave scanners began in the late 1970s with the Scanning Multichannel Microwave Radiometer (SMMR) and continued with the Special Sensor Microwave/Imager (SSM/I). In the past decade with the launch of Advanced Microwave Scanning Radiometer (AMSR) there is a decade long dataset (2002-2011) of SM estimates from space, and the effort continued with the European Space Agency Soil Moisture and Ocean Salinity Mission (SMOS). Recently developed observations from Soil Moisture Active Passive (SMAP) mission (Entekhabi

et al., 2010) provides a great opportunity in evaluating our ability to predict/forecast SM conditions, becasue of its superior quality compared to other satellite sensors (Chen et al., 2018). Thus, this study is motivated by exploiting SMAP data to validate Monthly-to-Seasonal soil moisture forecasting. SMAP being an L-band sensor has a deeper penetration depth, hence a higher sensitivity to moisture content in the top layer of soil. Also SMAP data are provided at a 36km resolution and resampled at 9km resolution where the latter resolution makes it very appropriate for our study. In addition, SMAP observations at 6am

and 6pm capture the significant time points of the diurnal hydrological cycle. (Entekhabi et al., 2010).

The main intent of this study is (1) to develop M2S SM forecasts from Noah3.2 LSM forced with climate forecasts and (2) to evaluate the skill of SM forecasts based on SM observations from SMAP satellite over the Southeast US (SEUS). To our knowledge, this is the first effort that evaluates the skill of a LSM in developing SM forecasts based on SMAP observations over a large region. We also evaluate SM forecasting methodology in predicting the severity of a historical drought happened

in 2007 over the study region. The next section briefly describes the data and forecasting methodology, followed by the results and evaluation of the forecasting skill and discussion.

## 2  Hydroclimatic Data and Methodology

This study utilizes Noah3.2 LSM to develop monthly SM simulations and M2S SM forecasts over the SEUS. *Noah LSM* has been developed from 1993 through multi-institutional cooperation and has been widely used in operational weather and

climate predictions (Ek et al., 2003). It also exhibits significant skill in developing monthly to seasonal streamflow forecasts over the study region (Mazrooei et al., 2015). The Noah3.2 LSM is executed within the NASA's Land Information System (LIS) framework (Kumar et al., 2006) designed for high performance hydrological modeling. Under the forecasting scheme, precipitation forecasts from ECHAM4.5 Atmospheric General Circulation Model (AGCM) along with the hourly climatology of non-precipitation meteorological forcing variables (e.g. wind speed, humidity, net SW/LW radiations, etc.) are used to

implement the LSM.

Phase 2 of the North American Land Data Assimilation System (NLDAS-2) is a comprehensive dataset available at relatively fine spatio-temporal resolution (hourly temporal scale and 1/8° spatial resolution) from 1979 to present (Mitchell et al., 2004). Hence, it provides a valuable basis to compute hourly climatological forcings for hydrologic forecasting purpose. Under the



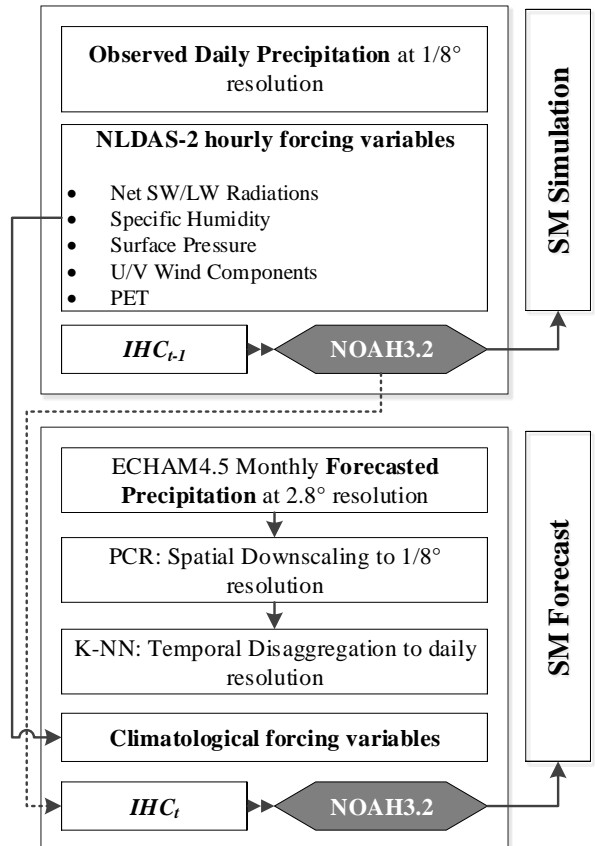

**Figure 1.** Flow chart of the Soil Moisture simulation and forecasting processes.

forecasting scheme, the hourly climatological forcings (i.e. hourly mean of NLDAS-2 forcings over a period of 31 years 1979-2010) are fed to the LSM.

Land-surface IHCs are one of the key components of LSMs in seasonal hydrologic forecasting where the predictability of the terrestrial fluxes is associated with the accuracy of the IHCs (Wood et al., 2016). In order to prepare adequate estimates of
5   IHCs prior to forecasting, NLDAS-2 meteorological forcings are used to run the Noah3.2 LSM in a simulation scheme (Figure 1). The computed hydrologic conditions at the end of the simulation period are then used to update the model's IHCs at the beginning of each forecasting period.

## 2.1   ECHAM4.5 Precipitation Forecasts

Besides the climatological forcings, precipitation forecasts from ECHAM4.5 AGCM are used in the forecasting approach.
10   ECHAM4.5 climate forecasts are more skillful than hourly climatology of NLDAS-2 precipitation variable because they inherit the ENSO signals (Mazrooei et al., 2015). ECHAM4.5 precipitation forecasts are obtained from the International Research


Institute for Climate and Society (IRI) Climate Data Library (Li and Goddard, 2005). These forecasts are available at 2.8°
spatial resolution and monthly time scales from January 1957 to present with lead times up to 7-months ahead consisting of 24
ensemble members. Constructed analogue Sea Surface Temperature (SST) forecasts have been used to develop the ECHAM4.5
AGCM climate forecasts. The spatial and temporal resolutions of the climate forecasts are much coarser than the resolution of
the Noah3.2 LSM forcing variables (i.e. 1/8°), thus statistical downscaling and disaggregation methods are employed in order
to address this mismatch.

Monthly precipitation forecasts are first spatially downscaled from 2.8° to 1/8° resolution through a Principal Component
Regression (PCR) model and then a Kernel Nearest Neighbor (K-NN) approach is applied in order to reproduce daily time-
series form a monthly forecast. For each 1/8° grid cell over the study region, four nearest 2.8° grid cells from ECHAM4.5
AGCM are identified as the PCR predictors and the observed monthly precipitation at 1/8° resolution from Maurer et al.
(2002) is used to train the PCR model. The PCR model is executed in a retroactive mode for each forecasting month (from
April 2015 to September 2016) using 54 years of data (from 1957 to 2010) as the training period. This time period is the
intersection of the intervals of the observational data and the ECHAM4.5 forecasts. For example, in order to obtain down-
scaled forecasts for January 2016, all the January data from 1957 to 2010 serves as the training dataset. Next, using the K-NN
disaggregation approach, the downscaled monthly forecast is compared to the historical observations of the same month (from
1949 to 2010) to identify and rank the nearest neighbors (i.e. months with the closest quantity). The observed daily precip-
itation corresponding to the identified months are resampled based on Lall and Sharma (1996) kernel. The K-NN temporal
disaggregation scheme preserves the monthly precipitation totals during the daily-resampling process. The explained steps are
applied to the ECHAM4.5 forecasts in order to develop 1-3 month ahead daily precipitation forecasts (Figure 1). Further details
of downscaling and disaggregation methods, the assessment of uncertainty propagation, and the seasonal skill of downscaled
precipitation forecasts can be found in Mazrooei et al. (2015).

Under the LSM forecasting mode (Figure 1), spatially downscaled and temporally disaggregated precipitation forecasts
along with the hourly climatology of the NLDAS-2 non-precipitation forcing variables are implemented to run Noah3.2 LSM
in 30-minute time steps. This setup is performed at the beginning of each month over the period February 2015 - September
2016 in order to develop up to 3-months ahead forecasts of hydrological fluxes. The Noah3.2 products are issued at daily time
scale and at 0.25° spatial resolution. Mean monthly SM forecasts of top 10cm layer of soil is computed by averaging daily
forecasted SM quantities.

## 2.2   SMAP Soil Moisture Data

The SMAP satellite was launched on January 31, 2015 designed to measure near surface (0-5 cm) SM and land surface
freeze/thaw conditions with a complete global coverage in 2-3 days (Entekhabi et al., 2010). In this study, Level-3 SMAP
radiometer global daily SM data at 9 Km spatial resolution is obtained from the National Snow and Ice Data Center (NSIDC)
(O'Neill et al., 2018). This data is available for the time period April 2015 to present of which we used the data over an
18-months period from April 2015 to September 2016.



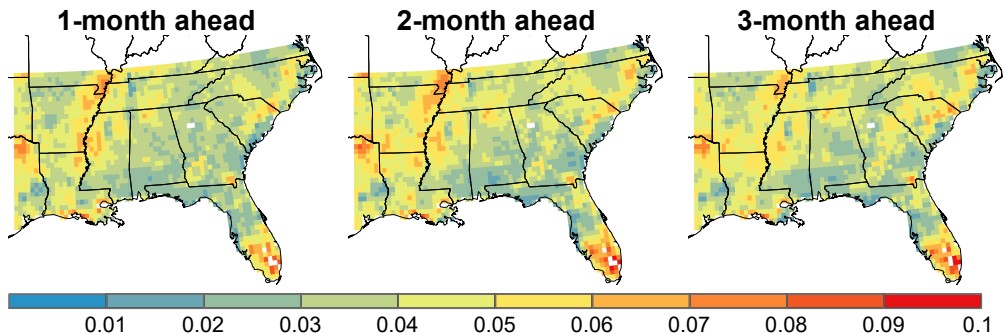

**Figure 2.** RMSE of the bias corrected 1-3 months ahead soil moisture forecasts based on the SMAP soil moisture observations

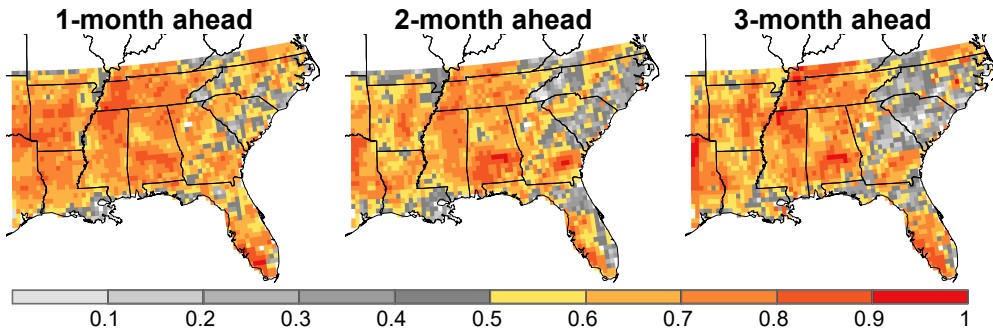

**Figure 3.** Correlation coefficient between 1-3 months ahead soil moisture forecasts with the SMAP soil moisture observations. grid cells with insignificant correlations (based on 18 monthly data points) are grayed out

To reproduce monthly SM observations matching spatio-temporal resolution of the LSM products, 9km daily observations during a specific month are averaged and upscaled to 0.25°. Given a 0.25° grid cell, the daily SMAP observations within a circular window circumscribed on the grid cell are averaged to represent the monthly observation for that location.

Furthermore, for each grid cell a uniform bias correction is applied to the timeseries of monthly SM forecasts from Noah3.2 LSM based on the difference between the mean of SMAP observations and the mean of forecasts over the 18-months study period. Monthly bias-corrected SM forecasts (in three different lead-times) are then compared to the corresponding monthly time-series of SMAP observations using correlation coefficient and Root Mean Squared Error (RMSE) metric in order to quantify the forecasting skill.

## 3 Results

Figures 2 and 3 show the RMSE and correlation coefficients between the bias-corrected monthly SM forecasts and monthly SMAP observations for 1-3 month lead times. Since 18 monthly values are used for the correlation quantification, grids with insignificant correlation coefficient at 95% confidence interval ($\pm 1.96/\sqrt{n}$, where $n$ denotes the length of data points) are





plotted in a gray scale (Steel et al., 1960). From Figure 2, higher RMSE occur over regions with predominantly wetland soil (e.g. Mississippi) and over regions with low content of clay abundant soil with slight swelling potential (e.g. eastern side of North Carolina and South Carolina states) according to Olive et al. (1989). The RMSE is also higher over the wetlands of the Everglades. The SM forecasts from LSM has lower RMSE and higher correlation over Alabama-Coosa-Tallapoosa (ACT),

Tennessee River Basins, and over the east flowing rivers of GA. SM forecasts also have limited skill over the western parts of NC and SC with the correlation becoming insignificant as a result of increasing forecast lead time.

Among all the 2121 grid cells covering our study domain, about 23% of the grid cells show a slightly increased RMSE due to a longer forecast lead time, mostly located in the southeast side of Appalachian mountains. Over most grid points, the forecasting error, RMSE, does not change significantly with increase in lead time, which indicates the strong role of IHCs

and limited skill of precipitation forecasts over the SEUS (Koster et al., 2010; Sinha et al., 2014). The spatially averaged RMSEs over the SEUS are equal to 0.039, 0.042 ,and 0.041 for 1-month, 2-month, and 3-month lead times respectively. The minimal change in RMSE across different lead times expresses the strong memory (persistence) of SM over SEUS. However, based on the correlation coefficients in Figure 3, when the lead time increases from 1 month to 3 months, number of grid cells with insignificant correlation increases specifically over the southern side of Appalachian. On the other hand, areas

with significant presence of deep soils (Effland, 2008) such as Mississippi, Alabama, and eastern side of Texas state indicate increased correlation coefficients in longer forecasting lead times. Along with the SM persistence, initializing the Noah3.2 LSM with simulated hydrologic conditions has a strong influence in improving the SM forecasting even for longer lead times (Shukla and Lettenmaier, 2011). The spatially averaged correlation coefficients are equal to 0.62, 0.57, and 0.58 for 1-3 month lead times respectively. Overall, the skill of the SM forecasts declines slightly with increasing lead-time due to the errors in

imprecise precipitation forecasts.

To further understand how the forecasts capture the variability in SM observations, two regions (each including four grid cells) with high and low skill in forecasting are selected and the anomalies around the mean of SM observations are presented in Figure 4. This figure also includes linear model fits and the prediction intervals at 95% confidence level. The first column shows scatter plots between the anomalies of the forecasts and the observations over four neighboring grid cells with relatively low

RMSE (0.019 on average) and a strong correlation coefficient (0.726 on average) located in Alabama state. The second column shows similar information from the pack of four grid cells located in South Carolina with poor forecasting skill (high RMSE and low correlations). The $R^2$ quantity included in each plot indicates the ability of forecasts in explaining the variability in SMAP observations, also the declining slope of the fitted line implies the increasing forecasting error for longer lead times.

As a test case, our SM forecasting methodology is examined during a historical drought event happened in Fall 2007 over

the study region. Since the SMAP data is not available for that time, the forecasted SM products are compared with the US Drought Monitor (USDM) data from National Drought Mitigation Center (Svoboda et al., 2002; Seager et al., 2009). For this purpose, we developed 1-month ahead SM forecasts over the region where the Noah3.2 model was initialized on Oct 1st 2007 using NLDAS-2 meteorological forcing variables. Then, for a given grid cell, the percentile of the SM forecast was computed from the monthly SM climatological distribution (for October) constructed based on a continuous 20-year

SM simulation using the Noah3.2 model (1991-2010). Thus, the SM forecasted percentiles indicate the likely deviations of

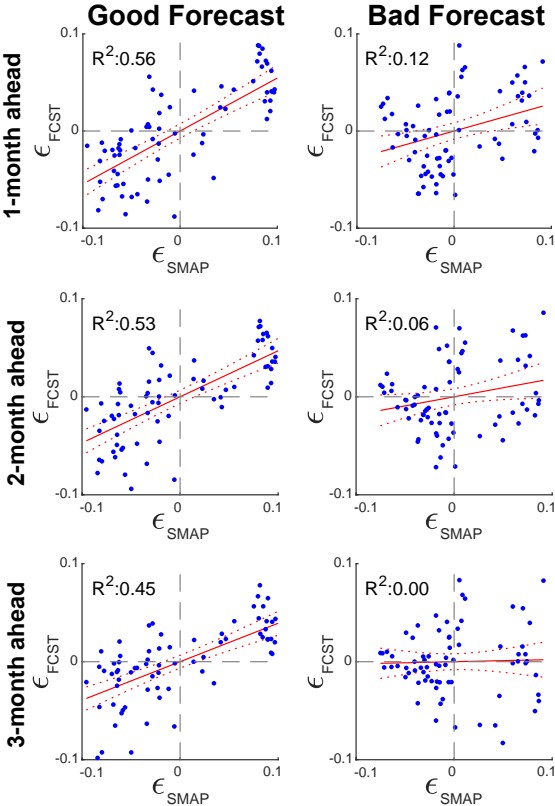

**Figure 4.** Scatter plots of soil moisture residuals for two sets of sample grid cells with good and bad forecasting skills. The residuals are centered around the mean of SMAP observations.

monthly SM from its climatological probability. Figure 5a demonstrates the forecasted percentiles of SM being less than 0.5 which identifies the SM below-normal conditions compared to the climatology. Comparing Figure 5a with the USDM drought monitoring conditions (obtained from droughtmonitor.unl.edu) issued on Oct 16[th] 2007 (Figure 5b), we see the 1-month ahead SM forecast captures the spatial pattern of the drought severity highlighting the hard-hit regions with lower percentiles of SM forecast. The promising skill in the SM forecasted percentiles provides a validation of the applied methodology for long-range forecasting. Furthermore, it exemplifies the value of long-range climate forecasts coupled with LSMs in developing real-time hydrologic forecasts in order to alleviate critical water-related issues.

## 4 Discussion

The main focus of this study is to develop monthly-to-seasonal (M2S) soil moisture (SM) forecasts through Noah3.2 LSM using ECHAM4.5 precipitation forecasts and evaluate the skill in SM forecasting by a comparison with the newly emerging SM observations from the SMAP satellite. Efforts have primarily focused on evaluating the skill of M2S SM forecasting over



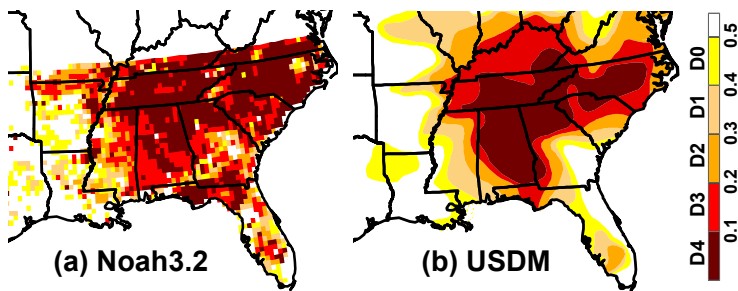

**Figure 5.** Comparison between the (a) Noah3.2 retrospective 1 month ahead soil moisture forecasted percentiles with (b) the actual drought conditions from U.S. Drought Monitor (USDM) during historical drought of Oct 2007

CONUS US by comparing with the model simulation driven by observed forcing as a benchmark (Mo et al., 2012; Mo and Lettenmaier, 2014). Integration of the ECHAM4.5 precipitation forecasts with the NLDAS-2 non-precipitation forcing variables supports the idea to evaluate the LSM in real-time SM forecasting. Our previous studies have also showed the robust performance of ECHAM4.5 forecasts for improving streamflow forecasting (Sinha et al., 2014; Mazrooei and Sankarasubra-

manian, 2017). Both forecast verification metrics, correlation coefficient and RMSE, show that the forecasted SM captures the variability in SMAP observations with decent accuracy. There is a slight skill reduction in SM forecasting as the forecasting lead time increases. Comparison with the reported USDM conditions also indicate the SM forecasts capture the spatial variability in drought conditions suggesting potential for developing real-time and practical M2S SM forecasts to inform dry/wet conditions in support of agricultural planning. To disseminate the forecasting approach with agencies, the hydroclimatology

group at North Carolina State University (NCSU) with collaboration of North Carolina state Climate Office have developed a SM and streamflow forecasting portal that automatically develops forecasts in real-time and updates the percentiles of SM forecasts by comparing it with the climatological distribution of long-term simulated SM (Arumugam et al., 2015). Most of the skill in SM forecasting is predominantly influenced by updating model initial conditions prior to forecasting. The skill of the SM forecasts also declines slightly with increasing lead-time due to the errors in imprecise precipitation forecasts. This has

been observed with streamflow forecasting where most of the skill in developing tercile streamflow forecasts primarily comes from updated initial conditions (Mazrooei and Sankarasubramanian, 2017, 2019).

Yet, the specification and quantification of different sources of uncertainty in SMAP data needs to be fully addressed to achieve a comprehensive assessment of forecasting skill. In addition, this study is limited using one particular GCM model for climate forecasts and one land surface model for hydroclimatic modeling. Hence, our findings can be expanded to future

research by examining and combining different LSMs and climate models. For instance, multimodel precipitation forecasts tend to improve the reliability of climate forecasts which could potentially improve the predictability of SM conditions. Moreover, the increasing availability of observational data from ongoing and future satellite missions along with the implementation of data assimilation methods would presumably improve the accuracy of the SM estimations and model's IHCs, and consequently increases the hydrologic forecasting skill (Liu et al., 2012).





**Author Contributions**

A.M. wrote the main manuscript text. A.M., S.A., and V.L. reviewed, and edited the manuscript. A.M. performed experiments and prepared figures. S.A. and V.L. designed the study.



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
