# Peer review of "Technical Note: Evaluation of the Skill in Monthly-to-Seasonal Soil Moisture Forecasting Based on SMAP Satellite Observations over the Southeast US"

_Hydrology and Earth System Sciences, 2019_

## Referee Comment (RC1) · Anonymous Referee #1 · 8 Sep 2019

I finished my review for the paper titled "Evaluation of the skill in monthly-to-seasonal soil moisture forecasting based on SMAP satellite observations over the southeast US" by Mazrooei et al. This study evaluated the seasonal surface soil moisture forecasting skill based on LSM and dynamical climate forecasts. The SMAP satellite soil moisture observations were used to assess the forecasting skills. Overall, I see the topic of this study fits the HESS journal well and this paper is well-written. I only have one major comment about the drought case study.

I am puzzled why the authors included a drought study here because they focused
on the surface soil moisture forecast. It is important to note that drought is a multi-faceted disaster and soil moisture can only partially characterize the agricultural drought (vegetation is another import factor). In particular, ROOT-ZONE SOIL MOISTURE should be used instead of SURFACE SOIL MOISTURE. Based on my limited literature review and publications reading, I did not see any important drought paper using the surface soil moisture to quantify drought. In Figure 5, I see a good consistency between the Noah 1-month forecasts and USDM products. However, I am not convinced by the drought severity classification. First, in my opinion, 20-year simulation is too short to estimate soil moisture percentile from a climatological point. Second, 0.5 percentile should be normal condition (i.e., think it in Z-statistics) instead of D0 drought condition. Note that in USDM, the drought severity is classified as: D0 (abnormally dry, percentile

---

## Referee Comment (RC2) · Raghavendra Jana (Referee) · 20 Sep 2019

Review of Technical Note titled "Evaluation of the Skill in Monthly-to-Seasonal Soil Moisture Forecasting Based on SMAP Satellite Observations over the Southeast US"

Recommendation: Minor revision. The manuscript presents a look into generating soil moisture forecasts on a monthly-to-seasonal basis, validated with SMAP data. The topic is interesting and suitable for HESS. The manuscript is well written for the most part. I have a few concerns that I lay out below: 1. P1, L15: Should be "terrestrial",

and "primarily". 2. P2, L1-2: Efforts to develop SM forecasts through LSMs have been compared with simulated SM from LSMs? Not sure what the authors intend to state here. Right now, it reads as if the simulated SM is compared with itself. 3. P2, L3-4: Are there not many published studies comparing simulated SM with remotely sensed data? Would they not count as "systematic evaluations"? 4. P2, L31: NLDAS-2 is a comprehensive dataset of what properties? 5. P3, L10: ECHAM forecasts are more "skillful"? I am not sure how forecasts can be skillful. They may be more accurate. The ECHAM algorithm may be more skillful at generating more accurate forecasts. 6. P4, L3: SST forecasts "constructed" from what? 7. P6, L1-6: This is surprising. It has been shown many times in the past that the spatial variability of SM reduces significantly at the wet end of the curve. Hence, I would expect the performance to be better over the wetlands. Any thoughts as to why this phenomenon is observed in your study? 8. I would prefer it if the Results and Discussion sections were merged, with the discussion happening immediately after each result is presented. The current split format makes me skip ahead to the Discussion after each result is presented, just to see what the authors make of it. 9. I am not really convinced that the drought section fits into this study. a. The rest of the study uses SMAP as the benchmark, this portion uses USDM data, which are probably not at the same resolution as the simulations. b. Further, no numbers are presented for this comparison. I am not sure the spatial patterns really match that well. For example, the Florida panhandle is shown to be extremely dry in the Noah plot, but USDM says different. At the very least, another map showing the difference between the Noah and USDM values would've given a better picture. c. Thirdly, using surface SM to forecast drought is making things too simplistic. There are various other factors that need to be taken into account. These thoughts lead me to recommend that the authors remove the drought study from this note.

---

## Author Comment (AC1) · 23 Oct 2019

I finished my review for the paper titled "Evaluation of the skill in monthly-to-seasonal soil moisture forecasting based on SMAP satellite observations over the southeast US" by Mazrooei et al. This study evaluated the seasonal surface soil moisture forecasting skill based on LSM and dynamical climate forecasts. The SMAP satellite soil moisture observations were used to assess the forecasting skills. Overall, I see the topic of this study fits the HESS journal well and this paper is well-written. I only have one major comment about the drought case study.

I am puzzled why the authors included a drought study here because they focused on the surface soil moisture forecast. It is important to note that drought is a multi-faceted disaster and soil moisture can only partially characterize the agricul-tural drought (vegetation is another import factor). In particular, ROOT-ZONE SOIL MOISTURE should be used instead of SURFACE SOIL MOISTURE. Based on my limited literature review and publications reading, I did not see any important drought paper using the surface soil moisture to quantify drought. In Figure 5, I see a good consistency between the Noah 1-month forecasts and USDM products. However, I am not convinced by the drought severity classification. First, in my opinion, 20-year simulation is too short to estimate soil moisture percentile from a climatological point. Second, 0.5 percentile should be normal condition (i.e., think it in Z-statistics) instead of D0 drought

condition. Note that in USDM, the drought severity is classified as: D0 (abnormally dry, percentile  $\leq$ 30%), D1 (moderate drought, percentile  $\leq$ 20%), D2 (severe drought, percentile  $\leq$ 10%), D3 (extreme drought, percentile  $\leq$ 5%), and D4 (exceptional drought, percentile  $\leq$ 2%). See: https://droughtmonitor.unl.edu/AboutUSDM/AbouttheData/DroughtClassification.aspx, and Table 1 in Svoboda et al. (2002). I recommend the authors only focused on surface soil moisture forecast and validation and remove the drought case study.

References: Svoboda, M., LeComte, D., Hayes, M., Heim, R., Gleason, K., Angel, J., ... & Miskus, D. (2002). The drought monitor. Bulletin of the American Meteorological Society, 83(8), 1181-1190.

Response: Thank you for your review and comments. We agree that analysing and forecasting drought conditions should not be performed solely by top-layer SM variables, but the main idea behind the presented case study was to exhibit the high compatibility/similarity between the forecasted percentiles and the monitored drought indexes during a historical severe drought in the region.

As it was suggested by you and the other reviewer, the material related to the case study assessment is now completely removed from the manuscript, and some minor edits are applied in order to maintain the flow of the text as it was.

---

## Author Comment (AC2) · 23 Oct 2019

Review of Technical Note titled "Evaluation of the Skill in Monthly-to-Seasonal Soil Moisture Forecasting Based on SMAP Satellite Observations over the Southeast US"

Recommendation: Minor revision. The manuscript presents a look into generating soil moisture forecasts on a monthly-to-seasonal basis, validated with SMAP data. The topic is interesting and suitable for HESS. The manuscript is well written for the most part. I have a few concerns that I lay out below:

1. P1, L15: Should be "terrestrial",and "primarily". Response: The typos are now corrected.

2. P2, L1-2: Efforts to develop SM forecasts through LSMs have been compared with simulated SM from LSMs? Not sure what the authors intend to state here. Right now, it reads as if the simulated SM is compared with itself.

Response: This is true, it is compared with the products of the same model. Most studies have evaluated the actual forecasts of terrestrial variables (e.g. SM, Streamflow) based on the model products/simulations, rather than actual observations. The difference is then the type of utilized

model forcings (either climate forecasts, or meteorological observations) to conduct these two experiments. This sentence is now rephrased in the manuscript to better convey the message.

Most efforts in developing SM forecasts through land-surface models (LSMs) have actually been compared to the model's SM products under a simulation scheme -using observation-based atmospheric forcings to execute the model- as opposed to actual SM observations (Mo et al., 2012; Mo and Lettenmaier, 2014). Nevertheless, systematic evaluation of our ability to

3. P2, L3-4: Are there not many published studies comparing simulated SM with remotely sensed data? Would they not count as "systematic evaluations"?

Response: There are studies that compare model SM simulations and SM observations, which estimates the errors associated with the "model structure" itself (Narasimhan et al. 2005;Hain et al. 2011). On the other hand, some studies have used model simulations as the benchmark to evaluate the SM forecasts (Mo et al. 2012, Mo and Letternmaier 2014). However, the focus of this section of our introduction is to highlight the need for evaluating "actual forecasts" based on actual observations over a large domain such as remotely sensed data from SMAP satellite, to be accounted as the "true value".

- Narasimhan, B., et al. "Estimation of long-term soil moisture using a distributed parameter hydrologic model and verification using remotely sensed data." Transactions of the ASAE 48.3 (2005): 1101-1113.
- Hain, Christopher R., et al. "An intercomparison of available soil moisture estimates from thermal infrared and passive microwave remote sensing and land surface modeling." Journal of Geophysical Research: Atmospheres 116.D15 (2011).
- Mo, Kingtse C., et al. "Do Climate Forecast System (CFSv2) forecasts improve seasonal soil moisture prediction?." Geophysical Research Letters 39.23 (2012).
- Mo, Kingtse C., and Dennis P. Lettenmaier. "Hydrologic prediction over the conterminous United States using the national multi-model ensemble." Journal of Hydrometeorology 15.4 (2014): 1457-1472.

**4. P2, L31: NLDAS-2 is a comprehensive dataset of what properties?**

Response: NLDAS-2 contains 11 fields of meteorological forcings in fine spatial resolution of 1/8° coverning U.S., favorable to conduct large-scale hydrological modelling. The content of NLDAS-2 dataset is now explained in the manuscript:

Phase 2 of the North American Land Data Assimilation System (NLDAS-2) is a comprehensive dataset of meteorological forcings available at relatively fine spatio-temporal resolution (hourly temporal scale and 1/8° spatial resolution) from 1979 to present (Mitchell et al., 2004). Hence, it provides a valuable basis to compute hourly climatological forcings for hydrologic

5. P3, L10: ECHAM forecasts are more "skillful"? I am not sure how forecasts can be skillful. They may be more accurate. The ECHAM algorithm may be more skillful at generating more accurate forecasts.

Response: We mean that ECHAM4.5 has skillful predictions of climate variables as opposed to the computed climatology based on long-term data. The term "skillful forecast" is a commonly used term in our publications (Mazrooei et al. 2015; Mazrooei and Sankar 2017;2019), which

infers that a selected methodology/algorithm outperforms another one in developing accurate forecasts.

- Mazrooei, Amirhossein, et al. "Decomposition of sources of errors in seasonal streamflow forecasting over the US Sunbelt." Journal of Geophysical Research: Atmospheres 120.23 (2015): 11-809.
- Mazrooei, Amirhossein, and A. Sankarasubramanian. "Utilizing probabilistic downscaling methods to develop streamflow forecasts from climate forecasts." Journal of Hydrometeorology 18.11 (2017): 2959-2972.
- Mazrooei, Amirhossein, and A. Sankarasubramanian. "Improving monthly streamflow forecasts through assimilation of observed streamflow for rainfall-dominated basins across the CONUS." Journal of Hydrology 575 (2019): 704-715.

**6. P4, L3: SST forecasts "constructed" from what?**

Response: The GCM climate model is forced with the updated SST forecasts -developed using the constructed analog SST(CA-SST, Van Del Dool 1994); rather than using observed SST- in order to force GCMs and develop climate forecasts.

Van den Dool, H. M. "Searching for analogues, how long must we wait?." Tellus A 46.3 (1994): 314-324.

7. P6, L1-6: This is surprising. It has been shown many times in the past that the spatial variability of SM reduces significantly at the wet end of the curve. Hence, I would expect the performance to be better over the wetlands. Any thoughts as to why this phenomenon is observed in your study?

Response: Since RMSE is quantified using squared differences, the forecasting skill tends to be overestimated over low SM quantities and underestimated over high SM quantities. Another appropriate approach to quantify and visualize the forecasting skill is to use Kling-Gupta Efficiency (KGE) score (Gupta et al., 2009), which simultaneously accounts for correlation coefficient, mean bias, and relative variability in the predictions and observations. This additional analysis is now presented in this discussion letter for a better comparison between wet/dry regions of our study.

*Gupta, Hoshin V., et al. "Decomposition of the mean squared error and NSE performance criteria: Implications for improving hydrological modelling." Journal of hydrology* 377.1-2 (2009): 80-91.

8. I would prefer it if the Results and Discussion sections were merged, with the discussion happening immediately after each result is presented. The current split format makes me skip ahead to the Discussion after each result is presented, just to see what the authors make of it. Response: Given that we have explained the methodology first, and then arranged the figures as a set of results, we would like to present them separately in the result section, followed by the discussion. We believe that the current flow of the manuscript is reasonable.

9. I am not really convinced that the drought section fits into this study. a. The rest of the study uses SMAP as the benchmark, this portion uses USDM data, which are probably not at the same resolution as the simulations. b. Further, no numbers are presented for this comparison. I am not sure the spatial patterns really match that well. For example, the Florida panhandle is shown to be extremely dry in the Noah plot, but USDM says different. At the very least, another map showing the difference between the Noah and USDM values would've given a better picture. c. Thirdly, using surface SM to forecast drought is making things too simplistic. There are various other factors that need to be taken into account. These thoughts lead me to recommend that the authors remove the drought study from this note.
Response: As it was suggested by you and the other reviewer, the material related to the case study assessment is now completely removed from the manuscript. We agree that analysing and forecasting drought conditions should not be performed solely by top-layer SM variables. The main idea behind the presented case study was to exhibit the high compatibility/similarity between the forecasted percentiles and the monitored drought indexes during a historical severe drought in the region.

Thanks for the detailed review and comments.

---

## Author Response (AR1)

Dear Miriam,

First, I want to thank you for handling our paper hess-2019-298 "Technical Note: Evaluation of the Skill in Monthly-to-Seasonal Soil Moisture Forecasting Based on SMAP Satellite Observations over the Southeast US". As you mentioned, I've revised the figure1 and its caption, now it includes an easy-to-read description of each component linked with the flowchart. Also a last paragraph is added to the discussion section describing the contribution of the paper and potential extension of the work. I've uploaded the new version of the paper in the HESS portal. Also a point-to-point comparison between old/new versions of the manuscript is attached to this letter for your consideration. The eliminated text from the old version is highlighted in red and the added text in the new version is highlighted in green:

Best regards,

Amir

**OLD VERSION**

[revised manuscript text omitted]

---

## Author Response (AR2)

Dear Miriam,

I've uploaded the new version of the paper in the HESS portal. To summarize the changes after minor revision: the title of last section of the manuscript is changed from "Discussion" to "Conclusion and Discussion". Also a paragraph is added at the end of this section that concludes the founding of our study. A point-by-point comparison between old/new versions of the last section of the manuscript is attached to this letter for your consideration. The eliminated text from the old version is highlighted in red and the added text in the new version is highlighted in green.

Best regards,

Amir Mazrooei; January 20, 2020

OLD VERSION:

[revised manuscript text omitted]